# Dosimetric Comparison of Conventional Radiotherapy, Volumetric Modulated Arc Therapy, and Proton Beam Therapy for Palliation of Thoracic Spine Metastases Secondary to Breast or Prostate Cancer

**DOI:** 10.3390/cancers15245736

**Published:** 2023-12-07

**Authors:** Anders Lideståhl, Emil Fredén, Albert Siegbahn, Gracinda Johansson, Pehr A. Lind

**Affiliations:** 1Department of Oncology-Pathology, Karolinska Institutet, 17177 Stockholm, Sweden; 2Department of Oncology, Stockholm South General Hospital, 11883 Stockholm, Sweden; emil.freden@regionstockholm.se (E.F.); albert.siegbahn@regionstockholm.se (A.S.); pehr.lind@regionstockholm.se (P.A.L.); 3Department of Clinical Science and Education, Karolinska Institutet, Stockholm South General Hospital, 17177 Stockholm, Sweden; 4Department of Radiotherapy, Uppsala University Hospital, 75185 Uppsala, Sweden; gracinda.johansson@akademiska.se

**Keywords:** spine metastases, thoracic, palliative radiotherapy, proton beam therapy, volumetric modulated arc therapy, planning study

## Abstract

**Simple Summary:**

Patients with breast or prostate cancer often develop painful tumors of the spine, i.e., spinal metastases. Treatment of these can be challenging as it is crucial to target the cancer cells without harming nearby healthy tissues. In this dose planning study, we compared three types of radiation therapy (RT) to see which could be most effective for treating spinal metastases while minimizing harm to nearby organs, such as the spinal cord, esophagus, heart, and lungs. Our results suggest that two advanced techniques, Volumetric Modulated Arc Therapy (VMAT) and Proton Beam Therapy (PBT), may be more precise in targeting the tumors and reducing potential harm to other organs compared to traditional palliative RT. These findings indicate that VMAT and PBT may offer better outcomes for certain groups of patients, but more research is needed to understand when these techniques could be appropriate.

**Abstract:**

The aim of this planning study was to compare the dosimetric outcomes of Volumetric Modulated Arc Therapy (VMAT), Proton Beam Therapy (PBT), and conventional External Beam Radiation Therapy (cEBRT) in the treatment of thoracic spinal metastases originating from breast or prostate cancer. Our study utilized data from 30 different treatment plans and evaluated target coverage and doses to vital organs at risk (OARs), such as the spinal cord, heart, esophagus, and lungs. The results showed that VMAT and PBT achieved superior target coverage and significantly lower doses to the spinal cord compared to cEBRT (target: median PTV_D95%_: 75.2 for cEBRT vs. 92.9 and 91.7 for VMAT (*p* < 0.001) and PBT (*p* < 0.001), respectively; spinal cord: median D_max%_: 105.1 for cEBRT vs. 100.4 and 103.6 for VMAT (*p* < 0.001) and PBT (*p* = 0.002), respectively). Specifically, VMAT was notable for its superior target coverage and PBT for significantly lower doses to heart, lungs, and esophagus. However, VMAT resulted in higher lung doses, indicating potential trade-offs among different techniques. The study demonstrated the relative advantages of VMAT and PBT over traditional RT in the palliative treatment of spinal metastases using conventional fractionation. These findings underscore the potential of VMAT and PBT to improve dosimetric outcomes, suggesting that they may be more suitable for certain patient groups for whom the sparing of specific OARs is especially important.

## 1. Introduction

Spinal metastases are common in advanced breast and prostate cancer [1] and lead to severe complications, e.g., pathological fractures, pain, and compressive myelopathy [2,3]. Many breast and prostate cancer patients (pts) with spinal metastases will undergo palliative radiotherapy (RT) during their illness [4]. Although stereotactic approaches may be indicated in specific circumstances (e.g., de novo metastases, oligometastases, and retreatment) [5], most pts are offered treatment with conventional fractionation regimens [6]. Commonly used regimens are 8 Gy in 1 fraction, 16 Gy in 2 fractions, 20 Gy in 5 fractions, and 30 Gy in 10 fractions. There are no confirmed differences in treatment outcomes such as pain relief or functional outcome between the regimens, although the need for retreatment seems to be higher after single-fraction treatments [7,8,9].

The standard RT technique for spinal metastases is conventional external beam radiotherapy (cEBRT) using simple static fields with posterior–anterior (PA) or two opposed anterior–posterior (AP-PA) photon beams [10]. The target volume in cEBRT is defined by 2D rectangular field borders and typically includes the affected vertebrae plus a safety margin of one extra vertebra above and below [11].

While cEBRT effectively treats pain and spinal cord compression in most pts with spinal metastases, the relief is often temporary, and up to 20% of pts need retreatment [9]. The radiation tolerance of the spinal cord limits treatment doses and consequently the possibility for retreatment [12]. In RT of thoracic spinal metastases, incidental radiation to other organs at risk (OARs), such as the heart, lungs, and esophagus, may also become dose-limiting [13].

Alternative RT techniques could be used in palliative RT with conventional fractionation to improve target dose coverage and minimize doses to the relevant OARs. Volumetric-Modulated Arc Therapy (VMAT) and Proton Beam Therapy (PBT) are two RT techniques with the potential to reduce doses to OARs and maintain adequate target coverage compared to cEBRT. VMAT delivers photon radiation in a continuous arc, while cEBRT employs one or multiple fixed beams [14]. Compared to photons, protons have a finite range which enables precise radiation delivery to tumors while limiting radiation exposure to OARs located distally to the target [15].

Primarily used in curative settings for various malignancies, VMAT and PBT are not commonly adopted in palliative RT due to more complex and time-consuming dose planning and, particularly in PBT, limited availability [16]. However, the use of VMAT and PBT in palliative RT of thoracic spinal metastases could theoretically reduce risks of acute and late radiation-induced complications, such as pneumonitis, esophagitis, pericarditis, and myelitis. Improved target dose coverage may potentially also reduce the need for retreatment.

The aim of this retrospective planning study was to compare cEBRT with simple static fields, with VMAT and PBT when treating pts with thoracic spinal metastases, with a focus on doses to the target and various OARs.

## 2. Materials and Methods

### 2.1. Patient and Treatment Characteristics

Twenty-nine pts with thoracic spine metastases who received palliative cEBRT at our institution were included in this study (Table 1). The cohort included 20 men with metastatic prostate cancer and 9 women with metastatic breast cancer. One patient was treated for two separate treatment targets, resulting in a total of 30 treatment plans for the entire patient cohort. The median age was 70 (range 40–89) years. The median number of vertebrae within the treatment target was 5 (range 3–13). In one case, two cervical vertebrae were also included in the treatment target. The indications for palliative RT were pain in 9 pts, threatening or manifest spinal cord compression in 16 pts, and postoperative treatment after palliative laminectomy in 4 pts. Doses and treatment regimens were as follows: 7 Gy × 1 (1 pt), 8 Gy × 1 (5 pts), 8 Gy × 2 (13 pts), 4 Gy × 5 (9 pts), and 5 Gy × 5 (1 pt).

### 2.2. Treatment, Setup, and Structure Delineation

A pre-treatment planning computed tomography (CT) scan in the supine position was acquired for all pts. We aimed to compare the actual treatments that the pts received with cEBRT to the alternative treatments with VMAT and PBT. In the original cEBRT treatment image sets, no CTV or PTV were delineated. Instead, the pts were treated with standard simple fields covering the affected vertebrae plus a margin encompassing an extra vertebra in both the cranial and caudal directions. For most cases, no extra margins were added in the lateral directions. To compare this treatment with alternative techniques (VMAT and PBT), CTV, PTV, and OARs were retrospectively delineated on the cEBRT dose planning CTs. Structure delineation was performed on the CT image sets by two oncologists (AL and PAL).

The clinical target volume (CTV) included the affected vertebrae in full and, in some cases, also the medial part of an affected costa (Figure 1). The planning target volume (PTV) was generated by applying a uniform margin of 0.5 cm in the anterior–posterior and lateral directions and a margin of 1 cm in the superior–inferior direction to the CTV. The OARs delineated were the heart, esophagus, lungs, and spinal cord (Figure 1). The esophagus was delineated from the cricoid to the gastroesophageal junction. The heart was delineated from the origin of the coronary vessels in the cranial direction down to the apex in the caudal direction and included the pericardium. The spinal cord encompassed the extension of the PTV in the axial direction, plus an extra margin of 2 cm in both the inferior and superior borders of the PTV. The lungs were delineated as a single OAR, i.e., the total lung volume (lung total). A representative CTV/PTV and OAR delineation is illustrated in the transversal and sagittal views in Figure 1 [patient #27].

### 2.3. Treatment Planning

Treatment plans with cEBRT were obtained from the clinical database, as these were the actual treatments administered to the pts. These plans were prepared on the Monaco™ (Elekta AB, Stockholm, Sweden) treatment planning system (TPS) with 1 or 2, 6 MV, and/or 15 MV posterior photon beams. In 24 of the treatment plans, only one dorsal field was used. In 6 of the treatment plans, 2 fields were utilized; in 5 of these plans, this configuration included an additional dorsal supplementary field, and in 1 of these plans, an additional anterior field was used to achieve more uniform dose coverage with depth across the entire target volume. The dose (to medium) calculation was performed with the Collapsed Cone algorithm with a dose grid of 0.3 cm and the treatments were delivered by a Versa HD™ (Elekta, AB, Stockholm, Sweden) linear accelerator in one to five treatment fractions, with fraction doses ranging from 4 Gy to 8 Gy (Table 1). The planning objective for the cEBRT plans was to cover as much as possible of the vertebrae anteriorly without exceeding the spinal cord constraint of 105% of the prescribed dose. Moreover, a global dose max constraint on the external body contour was set to 130% of the prescribed dose. 

All pts were retrospectively replanned with VMAT and PBT. The same OAR dose constraints used in the clinical cEBRT plans were adopted in both the VMAT and PBT plans. In addition, the planning objective for the VMAT and PBT plans was to cover 95% of the target volume with 90% of the prescribed dose, i.e., D95 ≥ 90%. For the VMAT plans, the objective was to cover the PTV in the nominal scenario, whereas in the PBT plans the objective was to cover the CTV in different treatment scenarios considering patient setup errors and range uncertainty. VMAT plans were prepared in Monaco™ (Elekta, AB, Stockholm, Sweden) TPS for treatment delivery with two full arcs. A Monte Carlo dose calculation algorithm was used, assuming dose deposition in water, with a statistical uncertainty of 1% and a dose grid of 0.3 cm.

The treatment planning for PBT was performed on the Eclipse™ treatment planning system (TPS) version 15 (Varian Medical Systems, Palo Alto, CA, USA). The planning for PBT was performed using spot-scanning beams with kinetic energies between 60 and 230 MeV generated in an IBA™ machine (Ion Beam Applications, S.A., Louvain-La-Neuve, Belgium). The plans were prepared using one field in a gantry position of 180 degrees. A range shifter of water with an equivalent thickness of 3.5 g/cm^2^ was used to ensure dose coverage of the part of the PTV located closer to the patient surface. A CTV-based robust optimization was performed, i.e., D95 ≥ 90% for the CTV, accounting for a setup uncertainty of 0.5 cm in the vertical and lateral direction and 1.0 cm in the axial direction, as well as a proton range uncertainty of 3.5%. For robustness evaluation purposes, we generated 12 treatment scenarios with perturbed dose distributions in addition to the unperturbed nominal plan. The dose calculation was performed using the proton convolution superposition algorithm (PSC), and the optimization was conducted with the Nonlinear Universal Proton Optimizer (NUPO) algorithm, both available in the Eclipse™ TPS. A relative biological effectiveness (RBE) of the proton beams of 1.1 was assumed.

### 2.4. Plan Evaluation

For both the cEBRT and the VMAT plans, the D_95_, D_2_, and D_mean_ of the PTV and the D_98_, D_2_, and D_mean_ of the CTV were registered. For the PBT plans, the D_98_, D_2_, and D_mean_ of the CTV were registered for the nominal plans, and the D_95_, D_2_, and D_mean_ of the CTV were registered for the worst-case scenario plans. For all OARs, the D_mean_, V_20_, V_30_, V_40_, and V_50_ were registered. For the spinal cord and the esophagus, the D_max_ was also registered. The dose–volume values obtained from the cEBRT plans were compared with the values obtained from both the VMAT plans and the nominal PBT plans. Since proton dose distribution is not shift-invariant in heterogeneous tissue, i.e., the static dose cloud approximation does not hold, it is difficult to directly compare PTV target coverage between photon and proton plans. Therefore, we mainly compared the CTV coverage in the worst-case proton scenario with PTV coverage in the cEBRT and VMAT plans. This approach is in accordance with previous recommendations [17]. For an easier comparison between different treatment regimens, all doses are presented as a percentage of the prescribed dose. Volume data are also presented using a relative dose, e.g., V_20%_ is the relative volume that receives 20% of the prescribed dose. Examples of typical dose distributions for the different treatment techniques are illustrated in Figure 2 (patient # 27).

### 2.5. Statistics

All variables were tested for normality and were found to be non-normally distributed. Therefore, a two-sided Wilcoxon signed-rank test was used to compare dosimetric outcomes amongst the cEBRT plans and the two alternative plans (VMAT and PBT) for each patient. A *p*-value < 0.05 was considered statistically significant. 

## 3. Results

Results are presented in Table 2 and Figure 3 and Figure 4. The target coverage planning objective (i.e., PTV_D95_ ≥ 90% for cEBRT and VMAT, and CTV_D95_ ≥ 90% for the worst-case scenario PBT plan) was met in 29/30 VMAT plans, in 28/30 PBT plans, and in 0/30 cEBRT plans. Target coverage was inferior (median PTV_D95%_: 75.2 for cEBRT vs. 92.9 and 91.7 for VMAT (*p* < 0.001) and PBT (*p* < 0.001), respectively), and doses to the spinal cord was higher in cEBRT compared to both VMAT and PBT (median D_max%_: 105.1 for cEBRT vs. 100.4 and 103.6 for VMAT (*p* < 0.001) and PBT (*p* = 0.002), respectively). Doses to the esophagus (e.g., median D_mean%_: 33.7 for cEBRT vs. 31.2 and 19.7 for VMAT (*p* = 0.001) and PBT (*p* < 0.001), respectively, medianV_30%_: 44.4 for cEBRT vs. 40.7 and 29.1 for VMAT (*p* < 0.001) and PBT (*p* < 0.001), respectively) and the heart (e.g., median D_mean%_: 23.4 for cEBRT vs. 13.4 and 0.68 for VMAT (*p* = 0.001) and PBT (*p* < 0.001), respectively, median V_30%_: 36.5 for cEBRT vs. 14.1 and 0.28 for VMAT (*p* < 0.001) and PBT (*p* < 0.001), respectively) were also significantly higher in cEBRT compared to VMAT and PBT, except for Dmax to the esophagus where cEBRT resulted in the lowest doses (median D_max%_: 85.1 for cEBRT vs. 103.9 and 107.4 for VMAT (*p* < 0.001) and PBT (*p* < 0.001), respectively). VMAT exhibited superior target coverage (median PTV_D95%_: 92.9 vs. 91.7 for VMAT and PBT, respectively (*p* = 0.001) and significantly lower doses to the spinal cord (D_max%_: 100.4 vs. 103.6 for VMAT and PBT, respectively, *p* < 0.001) compared to PBT. Lung doses were significantly higher in VMAT compared to both cEBRT and PBT (e.g., median D_mean%_: 17.9 for VMAT vs. 8.4 and 2.6 for cEBRT (*p* = 0.001) and PBT (*p* < 0.001), respectively, median V_30%_: 20.6 for VMAT vs. 8.4 and 3.3 for cEBRT (*p* < 0.001) and PBT (*p* < 0.001), respectively). PBT produced significantly lower doses to the heart, lungs, and esophagus compared to both VMAT (e.g., heart: median D_mean%_: 0.68 vs. 13.4 for PBT and VMAT, respectively, *p* < 0.001, median V_30%_: 0.28 vs. 14.1 for PBT and VMAT, respectively, *p* < 0.001; lung total: see above; esophagus: median D_mean%_: 19.7 vs. 31.2 for PBT and VMAT, respectively, *p* < 0.001, medianV_30%_: 29.1 vs. 40.7 for PBT and VMAT, respectively, *p* < 0.001) and cEBRT (e.g., heart: see above; esophagus: see above; lung total: median D_mean%_: 2.6 vs. 8.4 for PBT and cEBRT, respectively, *p* < 0.001, median V_30%_: 3.3 vs. 8.4 for PBT and cEBRT, respectively, *p* < 0.001).

## 4. Discussion

In our dose planning study, we demonstrated the dosimetric advantages of employing VMAT and PBT over cEBRT with simple static fields in palliative radiotherapy with conventional fractionation for thoracic spine metastases. The benefit of using VMAT or protons in our study, compared to cEBRT, comes from the fact that these methods first enable a more conformal dose distribution around the target, and second that the target volume was accurately delineated. cEBRT, under these circumstances, resulted in drastically inferior target coverage and significantly higher doses to the spinal cord compared to both alternative techniques. Thus, with cEBRT photon therapy with simple static fields, it is difficult to achieve sufficient coverage to the anterior part of the vertebra while avoiding excess doses to the spinal cord. Also, as lateral margins are usually not used in cEBRT, it may be difficult in some cases to cover the lateral aspects of the target. Both these factors contributed to inferior target coverage using cEBRT. The significant advantage of VMAT over cEBRT regarding target coverage lies in better dose coverage in the anterior parts of the vertebra (as illustrated in Figure 2). However, most thoracolumbar spinal metastases are predominately located in the posterior or middle–posterior part of the vertebra [18], suggesting that in some cases, limited coverage of the anterior part of the target using cEBRT may not be a major problem. Although VMAT showed superior target dose coverage and lower doses to the spinal cord compared to PBT, these differences may not be large enough to be of clinical importance. Conversely, the reduction in doses to the lungs, esophagus, and, especially, to the heart in PBT compared to both VMAT and cEBRT was of a magnitude that may be of clinical significance. For VMAT, the gain in target dose coverage, thus, came at the expense of considerably higher lung doses in comparison to both cEBRT and PBT due to unavoidable low-dose exposure to surrounding normal tissues with this technique. In cases at risk of intolerable doses to the lungs with VMAT, planning, including optimization for lung doses, should be considered. 

With the introduction of effective palliative systemic treatments, e.g., CDK4/6-inhibitors in breast cancer, and enzalutamide and abiraterone in prostate cancer, pts with bone metastases originating from these malignancies may have a life expectancy of several years [19,20]. Thus, the rise in the number of breast and prostate cancer pts with spine metastases who may not be eligible for retreatment with cEBRT due to excessive doses to OARs highlights a group of pts that could benefit from alternative RT techniques. For example, in breast cancer pts previously treated with cardiotoxic agents such as anthracyclines and left-sided adjuvant breast radiotherapy, the need to avoid further damage to the heart may motivate the use of VMAT or PBT, even in the palliative setting. Our study showed negligible doses to the heart with PBT when compared to VMAT and cEBRT, which makes PBT potentially more suitable for these pts.

In pts with a short life expectancy, treatment may be considered even when doses to OARs exceed their tolerance threshold. However, it is especially important to avoid severe acute radiation-induced complications, such as acute pneumonitis and esophagitis, which could significantly reduce the quality of life in this group of pts. Therefore, in pts with a short life expectancy, it is likely most important to avoid high doses to the esophagus and large lung volumes. Conversely, in pts with a longer life expectancy, i.e., several years, it is probably more crucial to avoid high doses to the spine and heart.

The choice of RT modality for the palliative treatment of spinal metastases should consider the balance between cost-effectiveness, treatment duration, and accessibility. cEBRT is both time- and cost-effective as it requires minimal contouring, dosimetric calculations, and quality assurance procedures [21]. Furthermore, it is widely available. VMAT is more time-consuming during the planning stage. A reduction in delivery time is, however, often highlighted as an advantage in favor of VMAT compared to 3D-CRT and IMRT [22], and this advantage may also be true compared to cEBRT treatments with multiple fields [23]. However, when compared to ‘simple’ cEBRT treatments, as in our study, VMAT offers no advantage regarding treatment delivery time (in our study, the number of monitor units was about 2.5 times larger for the VMAT plans). We created VMAT plans with flattened beams and 2 arcs, but single-arc and flattening filter-free plans can be used to shorten treatment times [24]. VMAT is less readily available than cEBRT but more accessible than PBT. PBT is the most expensive option among the three, primarily due to the complex infrastructure and technology required; therefore, PBT facilities are limited and concentrated to highly specialized centers. The current international trend is to construct more PBT facilities, which may eventually increase its availability for palliative patients. However, due to its high cost and complex delivery, PBT may remain suitable for only a defined subgroup of palliative patients for whom avoiding long-term toxicity is crucial. It is noteworthy that from the perspective of PBT providers, increasing the number of ‘simple’ treatments, such as palliations, may be of interest to improve profitability [25]. In conclusion, cEBRT has many advantages in terms of cost-effectiveness, treatment duration, and accessibility compared to VMAT, and especially PBT, in the palliative treatment of spinal metastases. However, one cannot rule out that the superior target coverage in VMAT and PBT may reduce the need for re-treatment, which would improve the cost-effectiveness of these modalities. Also, a reduction in acute and late radiation-induced complications with VMAT and PBT should further improve the economy of using these techniques.

A limitation of this study lies in not comparing VMAT and PBT with optimized cEBRT treatments. Thus, CTVs, PTVs, and OARs were retrospectively delineated on the original cEBRT dose planning CT sets, but the cEBRT treatments were not replanned using these volumes. However, the aim of our study was not to optimize cEBRT treatment but to compare the actual dosimetric outcomes of the standard treatment at our institution with those of VMAT and PBT. Had we optimized the cEBRT plans for target coverage, we would have ended up comparing three plans, none of which were used for actual treatment. Hence, we believe it was reasonable to ensure that the standard treatment was accurately described.

For the calculation of the RBE-weighted dose, we adopted a constant RBE = 1.1 as per clinical routine. However, a concern worth mentioning is the uncertain/increasing RBE at the end of proton tracks [26] which, in combination with an uncertain proton range [27], may lead to unacceptably high biological doses in critical structures located in the vicinity of the target. In our study, where we used a single 180-degree dorsal field, these end-of-range effects increase the uncertainty of the RBE-weighted dose delivered to the spinal cord (which is located within the target volume), as well as to the esophagus and heart (located anterior to the target volume). As the esophagus is sensitive to receiving high doses even to small volumes, a safer approach (considering these end-of-range effects) could be to redirect the beam away from the esophagus and place the distal edge of the beam in the lungs instead, as high doses to small volumes of the lung should be less detrimental due to its parallel architecture. To test whether different beam angles might improve target uniformity and decrease the dose to the spinal cord, as well as mitigate the end-of-range effect on the esophagus, without causing excessively high lung doses, two alternative PBT plans were created for patient # 27 (Figure 5). The first alternative plan was an intensity-modulated proton therapy (IMPT) plan with two posterior oblique beams using multi-field optimization (MFO) (2 post-obl-fields). The chosen beam angles were 125° and 235°. The second alternative plan used only one posterior oblique beam with a 125° angle (1 post-obl-field). These plans were compared to the original PBT plan (one posterior beam with a 180° angle) (1 dorsal field). Target coverage was similar between the plans (CTV (D_mean%_): 100.2%, 100.2%, and 97.6% for 1 dorsal field, 2 post-obl-fields, and 1 post-obl-field, respectively). Esophagus (D_max%_) was similar between the plans; however, esophagus (D_mean%_) was lower for the plan with one posterior oblique field (26.8%, 28.3%, and 15.6% for 1 dorsal field, 2 post-obl-fields, and 1 post-obl-field, respectively). Doses to the spinal cord were similar between the plans. Lung doses increased in both alternative plans (lung total (D_mean%_): 6.4%, 11.3%, and 14.6% for 1 dorsal field, 2 post-obl-fields, and 1 post-obl-field, respectively). Hence, for this patient, we could not observe any clear advantage of using different beam angles compared to the original PBT plan. Additionally, using oblique beam angles may have some potential drawbacks. First, the distance to the target increases, resulting in a longer travel range for the protons. Second, there is a longer stretch of lung volume, which may lead to dose perturbations due to tissue heterogeneity interfaces. Both these circumstances may increase range uncertainties in the dose calculations. It is worth noting, however, that this patient, like all patients in the study, was placed in a supine position with their arms along their sides. With their arms in this position, they obstruct the ability to use a shallower angle of incidence on the treatment field. One could consider a different setup with the arms over the head to achieve a shallower angle of incidence from the patient’s right side, which could help avoid the esophagus even with this field. Therefore, it is possible that a different patient setup could have achieved more effective sparing of the esophagus with a plan using two posterior oblique fields than was demonstrated in our patient example.

End-of-range effects can be considered during plan optimization through the implementation of variable RBE models or proton track-end objectives to reduce the RBE uncertainty within critical structures [28]; however, this was not possible in the TPS used in our study. 

Another aspect worth addressing is the robustness of different treatment techniques. For cEBRT plans with static rectangular fields, the delivered dose is not very sensitive to intra-fractional motion since the photon fluence is uniform and does not change over time. For VMAT treatments, on the other hand, for which the treatment geometry and dose rate are dynamically changing, intra-fractional motion could lead to interplay effects with local under- or overdosage [28]. Motion during PBT spot scanning treatments could also lead to similar interplay effects since each spot is delivered sequentially [28].

There are few available studies on VMAT or PBT in RT using conventional fractionation regimens for spine metastases as most studies focus on stereotactic approaches. In a planning study by Rief and colleagues, a 24 Gy single-fraction spinal SBRT treatment, performed with intensity modulated RT (IMRT), in 3 pts with cervical, thoracic, or lumbar bone metastases, was compared to alternative single-fraction carbon ion RT and PBT plans [29]. There were no significant differences in PTV coverage between the different techniques. For the thoracic lesion, there was no significant difference in dose to the spinal cord between the different techniques. The D_10_ for the esophagus was significantly lower in PBT compared to both IMRT and carbon ion RT (0.1 Gy vs. 5.6 Gy and 1.0 Gy, respectively). Lung doses were significantly higher with photons compared to both the alternative techniques. 

Remberg Gram and co-workers demonstrated the feasibility of VMAT in sparing the esophagus in a planning study on 29 pts with metastatic spinal cord compression (MSCC) [30]. In an ongoing clinical trial, a prospective cohort of 65 pts with MSCC is being treated with VMAT (or SBRT in pts with single spinal metastases) with 18 × 2.33 Gy [31]. The primary endpoint is local progression-free survival at 12 months, and the prospective cohort will be compared to a historic cohort of 235 pts treated with cEBRT (30 Gy in 10 fractions).

Two recent randomized studies that compared SBRT with conventional fractionation regimens for pain relief in pts with spinal metastases show contradicting results. In the study by Ryu et al., 339 pts with 1–3 sites of spinal metastases were randomly assigned to either receive stereotactic radiosurgery (SRS) with a single dose of 16–18 Gy or to conventional fractionation with 8 Gy × 1. The primary endpoint of patient-reported pain at 3 months was not superior in the SRS group [32]. In the second study by Saghal et al., 229 pts with spinal metastases were randomized to receive either SBRT (24 Gy in 2 fractions) or conventional fractionation (20 Gy in 5 fractions). There was a significant difference in the proportion of complete pain response at 3 months after RT in favor of SBRT compared to conventional fractionation (40% vs. 14%, respectively) [33]. Thus, at least for pain relief, which is the most common indication for palliative RT of spine metastases, stereotactic approaches may warrant further evaluation. Hence, RT with conventional fractionation should still be considered the gold standard [10]. This emphasizes the importance of investigating how palliative RT with conventional fractionation can be improved using alternative techniques.

## 5. Conclusions

In conclusion, this planning study indicates a potential role of VMAT and PBT, and target delineation, for improving target coverage and reducing doses to relevant OARs in the palliative treatment of pts with thoracic spinal metastases compared with cEBRT with simple static fields. Though standard cEBRT may be sufficient for most pts, specific circumstances, e.g., pts with known heart or lung conditions, could warrant the use of alternative techniques such as VMAT or PBT.

Additional research is necessary to evaluate the clinical importance of these dosimetric advantages on patient-reported outcomes and treatment-related toxicity. Also, an assessment of the cost-effectiveness of implementing VMAT and PBT for the palliative treatment of thoracic spinal metastases would be beneficial in guiding the decisions of healthcare providers and policymakers.

## Figures and Tables

**Figure 1 cancers-15-05736-f001:**
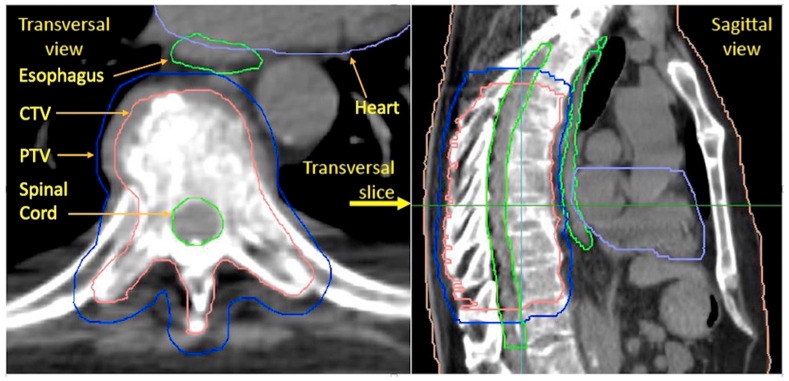
Example of typical delineations of CTV, PTV, esophagus, heart, and spinal cord (patient # 27). Note that the CTV includes the medial part of the affected left costa.

**Figure 2 cancers-15-05736-f002:**
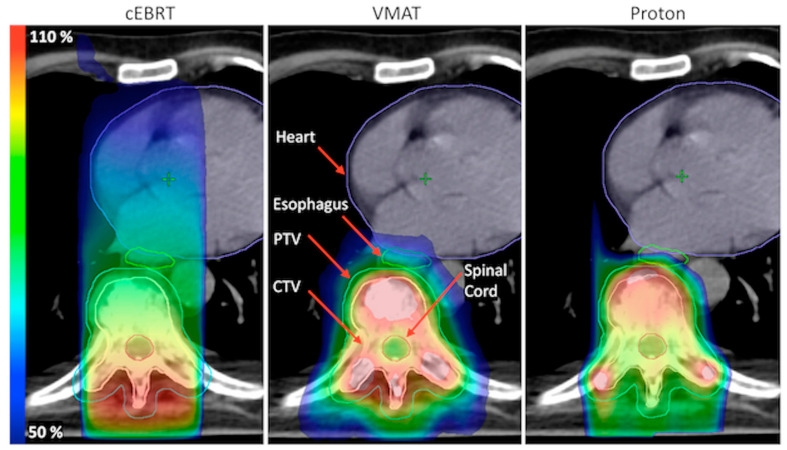
Dose distribution for patient # 27 comparing cEBRT, VMAT, and PBT. Note that the dose is lower in the anterior part of the PTV for cEBRT and that small parts of the lateral aspects of the PTV are underdosed.

**Figure 3 cancers-15-05736-f003:**
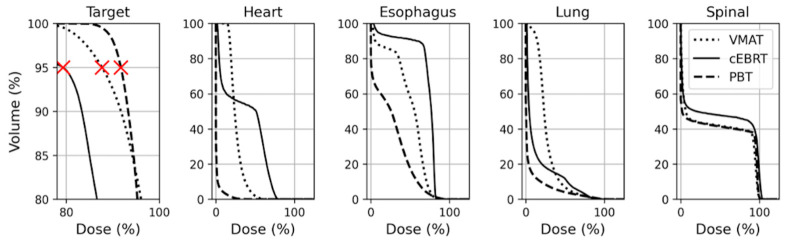
Dose volume histograms (DVHs) for PTV, heart, esophagus, lungs, and spinal cord comparing the different treatment techniques (patient # 27). The target coverage planning objective (i.e., PTV_D95_ ≥ 90% for cEBRT and VMAT, and CTV_D95_ ≥ 90% for the worst-case scenario PBT plan) is indicated with red crosses (Target). For this patient, the planning objective is met in only the PBT plan.

**Figure 4 cancers-15-05736-f004:**
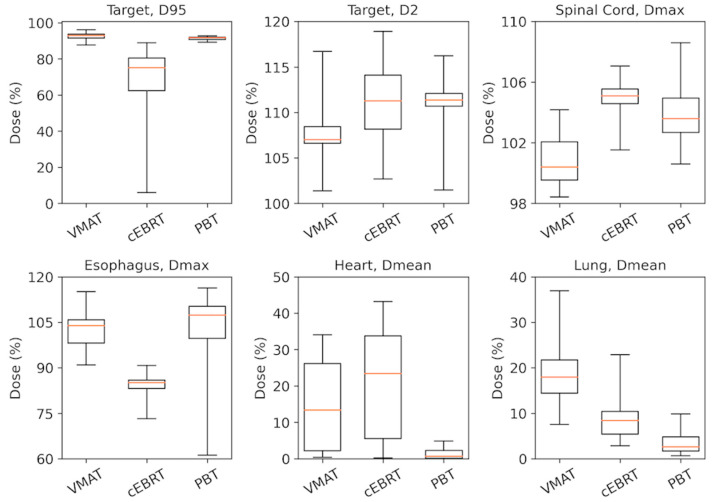
Box plots comparing PTV (D_95_), PTV (D_2_), spinal cord (D_max_)_,_ esophagus (D_max_)_,_ heart (D_mean_)_,_ and lung (D_mean_) between the different treatment techniques.

**Figure 5 cancers-15-05736-f005:**
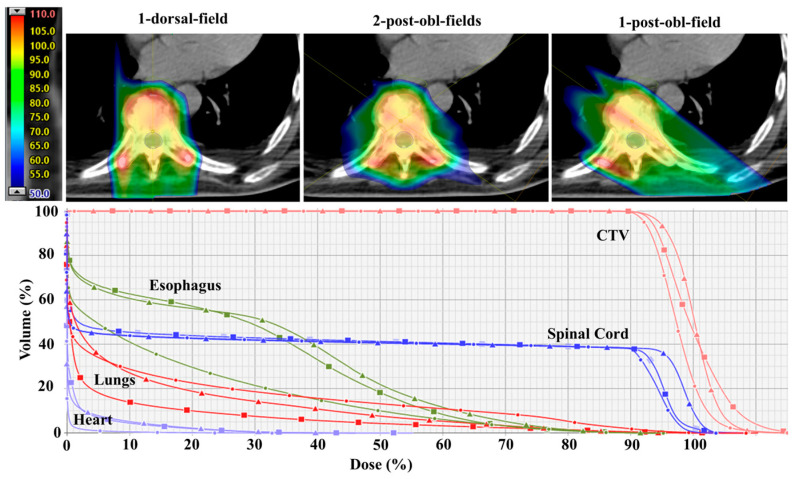
Upper row: dose distribution for patient # 27 comparing 1 dorsal field, 2 posterior oblique fields, and 1 posterior oblique field PBT plans. Lower row: DVH for the 3 different PBT plans. Key to symbols: square is 1 dorsal field, triangle is 2 posterior oblique fields, and circle is 1 posterior oblique field.

**Table 1 cancers-15-05736-t001:** Patient and treatment characteristics for the cEBRT plans.

Patient	Target	No. of Vertebrae within Target	RT-Regimen
1	T4-T8	5	8 Gy × 1
2	T3-T12	10	8 Gy × 2
3	T5-T7	3	8 Gy × 2
4	T2-T5	4	7 Gy × 1
5	T2-T11	10	4 Gy × 5
6	T3-T8	6	8 Gy × 2
7	T5-T9	5	8 Gy × 2
8	T1-T5	5	8 Gy × 1
9	T5-T9	5	4 Gy × 5
10	T7-T9	3	8 Gy × 2
11	T5-T8	4	4 Gy × 5
12	T4-T7	4	8 Gy × 2
13	T5-T8	4	4 Gy × 5
14	T5-T10	6	4 Gy × 5
15	T2-T8	7	5 Gy × 5
16	T4-T7	4	8 Gy × 2
17	T3-T8	6	4 Gy × 5
18	T7-T10	4	8 Gy × 2
19	T5-T8	4	8 Gy × 2
20	T2-T4/ T8-10	6	4 Gy × 5
21	T2-T4	3	8 Gy × 2
22	T7-T9	3	8 Gy × 1
23	T2-T6	5	4 Gy × 5
24	T2-T6	5	8 Gy × 2
25	T6-T9	4	8 Gy × 2
26	T1-T6	6	4 Gy × 5
27	T5-T7	3	8 Gy × 1
28	T7-T11	5	8 Gy × 2
29	C6-T11	13	8 Gy × 1

**Table 2 cancers-15-05736-t002:** Doses to targets and OARs. All doses are presented as a percentage of the prescribed dose. Volume data are also presented for the relative dose. Data are presented as median + IQR.

Target/OAR	Variable (%)	VMAT	cEBRT	PBT ^1^	*p*-Value		
					VMAT vs. cEBRT	VMAT vs. PBT	cEBRT vs. PBT
CTV	D_98_	93.4 (92.7–94.1)	82.0 (79.9–84.0)	92.3 (91.7–93.0)	<0.001	<0.001	<0.001
	D_2_	107.6 (107.0–109.0)	109.0 (106.8–111.5)	110.3 (109.5–111.0)	0.254	0.006	0.116
	D_mean_	101.8 (101.1–103.2)	95.4 (94.2–96.5)	100.9 (100.5–101.2)	<0.001	0.006	<0.001
PTV ^2^	D_95_	92.9 (91.6–93.7)	75.2 (60.6–81.0)	91.7 ^2^ (90.8–92.0)	<0.001	0.001	<0.001
	D_2_	107.0 (106.6–108.6)	111.3 (108.0–114.4)	111.4 ^2^ (110.7–112.1)	0.001	<0.001	0.829
	D_mean_	100.5 (99.5–101.3)	93.2 (90.8–95.2)	99.4 ^2^ (98.6–99.6)	<0.001	0.001	<0.001
Spinal Cord	D_max_	100.4 (99.5–102.2)	105.1 (104.6–105.6)	103.6 (102.6–105.0)	<0.001	<0.001	0.002
	D_mean_	78.5 (74.4–82.6)	85.3 (79.3–90.6)	75.2 (68.7–78.7)	0.001	<0.001	<0.001
Esophagus	D_max_	103.9 (97.7–106.1)	85.1 (83.2–86.0)	107.4 (99.4–110.7)	<0.001	0.043	<0.001
	D_mean_	31.2 (23.9–41.4)	33.7 (28.5–41.4)	19.7 (12.4–29.1)	0.001	<0.001	<0.001
	V_20_	42.5 (30.5–55.1)	45.8 (37.6–56.8)	32.1 (24.5–47.1)	<0.001	<0.001	<0.001
	V_30_	40.7 (30.5–53.6)	44.4 (36.0–55.5)	29.1 (19.9–44.7)	<0.001	<0.001	<0.001
	V_40_	36.9 (29.0–52.0)	43.9 (34.4–54.4)	25.5 (14.5–38.4)	<0.001	<0.001	<0.001
	V_50_	32.4 (26.4–44.7)	42.6 (33.2–53.1)	21.4 (10.2–32.3)	<0.001	<0.001	<0.001
Heart	D_mean_	13.4 (2.0–26.6)	23.4 (4.2–34.6)	0.68 (0.10–2.4)	<0.001	<0.001	<0.001
	V_20_	31.5 (0.23–66.7)	39.5 (5.3–58.3)	0.97 (0.00–4.3)	0.831	<0.001	<0.001
	V_30_	14.1 (0.00–38.1)	36.5 (4.5–56.0)	0.28 (0.00–2.5)	<0.001	<0.001	<0.001
	V_40_	6.0 (0.00–14.7)	32.7 (3.8–54.1)	0.09 (0.00–1.5)	<0.001	<0.001	<0.001
	V_50_	2.0 (0.00–6.6)	24.7 (2.5–44.6)	0.02 (0.00–0.83)	<0.001	<0.001	<0.001
Lung Total	D_mean_	17.9 (14.0–22.3)	8.4 (5.3–10.7)	2.6 (1.7–4.9)	<0.001	<0.001	<0.001
	V_20_	41.0 (31.4–55.1)	10.3 (6.8–14.1)	4.1 (2.5–7.4)	<0.001	<0.001	<0.001
	V_30_	20.6 (15.1–26.5)	8.4 (5.3–12.0)	3.3 (1.8–6.0)	<0.001	<0.001	<0.001
	V_40_	11.0 (7.9–14.0)	7.1 (4.3–9.8)	2.6 (1.4–4.8)	<0.001	<0.001	<0.001
	V_50_	6.8 (4.6–8.8)	4.8 (2.8–7.8)	2.1 (1.0–4.0)	0.039	<0.001	<0.001

^1^ nominal plan ^2^ for PBT, the worst-case scenario CTV is used for comparison with PTV for cEBRT and VMAT.

## Data Availability

The data presented in this study are available on request from the corresponding author. The source data are not publicly available as they are collected from the treatment planning system at the Department of Oncology at Stockholm South General Hospital, Stockholm, Sweden.

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
