# Peer review of "Dosimetric Comparison of Conventional Radiotherapy, Volumetric Modulated Arc Therapy, and Proton Beam Therapy for Palliation of Thoracic Spine Metastases Secondary to Breast or Prostate Cancer"

_cancers, 2023, doi:10.3390/cancers15245736_

Round 1
Reviewer 1 Report
Comments and Suggestions for Authors
First of all, it is hard to understand why the authors limited spinal metastases to prostate cancer and breast cancer and presented it in the title and introduction. Although the patient groups identified by the author are prostate and breast cancer patients, I do not think there is a need to limit the study since we are ultimately comparing treatment methods for spinal metastases. Ultimately, when comparing doses in treatment methods, the location of the primary cancer is not relevant.
Without CTV and PTV, cEBRT plan was constructed, and then the authors delineated CTV and PTV and generated VMAT and proton plan to cover targets, so, those plan results are shown to be superior to cEBRT plan. The authors presented the limitation of this study as not using an optimized cEBRT plan, to compare dosimetric benefits for avoiding severe acute radiation-induced complications (which are the benefits of using VMAT or proton therapy as the authors mentioned in the manuscript). However, at least target coverages of plans need to be optimized and then the authors could compare OAR dosimetric benefit.
Also, there are previous publications regarding to dosimetric advantage of VMAT and proton therapy for spine metastases radiotherapy.
Moveover, clinical outcome, planning time and resources, treatment time (because of patients’ pain), and cost-effectiveness are better to be included to convince clinical benefit of alternative techniques along with simple dosimetric comparison.
Reviewer 2 Report
Comments and Suggestions for Authors
I am grateful for the opportunity to review manuscript with ID: cancers-2657307, entitled “Dosimetric comparison of conventional radiotherapy, VMAT, and proton beam therapy for palliation of thoracic spine metastases secondary to breast or prostate cancer”.
With chemo-radiotherapy regimens prolonging the life of prostate and breast cancer patients with spine mets, investigating advanced techniques to improve radiotherapy for spine mets is a worthwhile exercise. The authors of this manuscript report their retrospective planning study comparing dosimetric parameters for VMAT and proton beam therapy (PBT) relative to conventional RT. A study cohort of 29 patients who previously received conventional RT to their spine mets was used. The original RT plan as delivered was maintained for the study. Target and OARs were drawn for the VMAT and PBT plans. The study showed that both VMAT and PBT achieved better target coverage and OAR sparing than conventional RT. VMAT showed slightly superior target coverage and cord sparing than PBT, whereas PBT significantly reduced doses to the lungs, esophagus, and heart.
The manuscript is well written; the data are clearly presented; the conclusions are supported by the data. The authors were also clear in describing the details of their VMAT and PBT planning approach, explaining the use of CTV-based robustness optimization for PBT. The discussion also addresses the issues of end-of-range LET effects on the RBE of the proton beams. This was something I was concerned about as I read the manuscript as the biological dose to the esophagus, for example, could be impacted by this. Overall this is a good paper that focusses on improving RT for thoracic spine mets. I have only one comment:
· The PBT plan uses pencil beam scanning (PBS). A single PA beam direction is used, and so the plan uses single-beam optimization (SFO), by definition. If the authors were to consider using two post-obl beam directions using multi-field optimization (MFO), aka IMPT, then perhaps the uniformity of target coverage could be improved, along with superior sparing of the cord. Furthermore, by using two pos-obl angles, the end-of-range effect on the esophagus could be mitigated somewhat. This may increase the lung dose, however, but perhaps not significantly. I suggest the authors at least generate a two-field IMPT plan for patient #27 (as this patient is used for the figures) to test this, and include this comparison as a discussion point.
Reviewer 3 Report
Comments and Suggestions for Authors
Good planning study of palliative RT for spinal metastases, clear and well written. Results expected based on planning techniques evaluated in this study. The main suggestion for improvement is to include a more robust discussion of the increased resources (time, money, availability) required for VMAT/PBT vs cEBRT. This is the primary reason why these more advanced techniques are not utilized more. Other minor suggestions for improvement:
Line 60 - clarify that cEBRT used only posterior/dorsal beams (no anterior beams)
Lines 130 and 189 - change 'pat' to full word 'patient'
Line 138 - as above, state specifically that all cEBRT used only posterior beams
Line 267 - 'clinical significance' has not yet been proven
Round 2
Reviewer 2 Report
Comments and Suggestions for Authors
With this revised submission, the authors have adequately addressed reviewer comments. I thank them for taking the time to investigate post-obl fields for patient 27.